

# A review of predator exclusion fencing to create mainland islands in Hawai'i

Lindsay Young and Eric VanderWerf

Pacific Rim Conservation, Honolulu, HI, United States of America

## ABSTRACT

**Background**. Invasive species are the primary threat to island ecosystems globally and are responsible for approximately two-thirds of all island species extinctions in the past 400 years. Non-native mammals—primarily rats, cats, mongooses, goats, sheep, and pigs—have had devastating impacts on at-risk species and are major factors in population declines and extinctions in Hawai'i. With the development of fencing technology that can exclude all mammalian predators, the focus for some locations in Hawai'i shifted from predator control to local eradication.

**Methods**. This article describes all existing and planned full predator exclusion fences in Hawai'i by documenting the size and design of each fence, the outcomes the predator eradications, maintenance issues at each fence, and the resulting native species responses.

**Results**. Twelve predator exclusion fences were constructed in the Hawaiian Islands from 2011–2023 and six more were planned or under construction; all were for the protection of native seabirds and waterbirds. Fences ranged in length from 304–4,877 m and enclosed 1.2–640 ha. One-third of the 18 fences were peninsula-style with open ends; the remaining two-thirds of the fences were complete enclosures. The purpose of twelve of the fences (67%) was to protect existing bird populations, and six (33%) were initiated for mitigation required under the U.S. Endangered Species Act. Of the six mitigation fences, 83% were for the social attraction of seabirds and one fence was for translocation of seabirds; none of the mitigation fences protected existing bird populations. Rats and mice were present in every predator exclusion fence site; mice were eradicated from five of six sites (83%) where they were targeted and rats (three species) were eradicated from eight of 11 sites (72%). Mongoose, cats, pigs, and deer were eradicated from every site where they were targeted. Predator incursions occurred in every fence. Rat and mouse incursions were in many cases chronic or complete reinvasions, but cat and mongoose incursions were occasional and depended on fence type (*i.e.*, enclosed *vs.* peninsula). The advent of predator exclusion fencing has resulted in great gains for protecting existing seabirds and waterbirds, which demonstrated dramatic increases in reproductive success and colony growth. With threats from invasive species expected to increase in the future, predator exclusion fencing will become an increasingly important tool in protecting island species.

Corresponding author
Lindsay Young,
lindsay@pacificrimconservation.org

## INTRODUCTION

Invasive non-native species are considered to be the second most important cause of global biodiversity loss (*Bellard, Cassey & Blackburn, 2015*; *Doherty et al., 2016*). These impacts are especially acute on oceanic islands where invasive species have been responsible for 86% of extinctions. On oceanic islands, many species evolved without mammalian predators and lack the appropriate responses and life history characteristics to avoid or repel them (*Vitousek, 1988*; *Blumstein & Daniel, 2005*). The number of eradications of invasive mammals from islands has increased in recent years, and these efforts have prevented several species extinctions and promoted spectacular recovery of native species (*Spatz et al., 2022*). However, island-wide eradication of invasive predators is not practical in some circumstances, and in those cases a combination of predator-exclusion fences and/or landscape level predator control programs have been used to create "mainland islands" to protect native species (*Saunders, 2001*; *Burns, Innes & Day, 2012*; *Dickman, 2012*; *Innes et al., 2019*; *Innes et al., 2024*). While early eradications and control programs focused on single species, recent efforts typically have targeted multiple species simultaneously (*Innes & Saunders, 2011*; *Lurgi, Ritchie & Fordham, 2018*). Conducting multi-species eradications often reduces costs (*Griffiths, 2011*) and prevents ecological release of smaller, non-target meso-predators (*Griffiths, 2011*; *Innes & Saunders, 2011*; *Lurgi, Ritchie & Fordham, 2018*).

The Hawaiian archipelago is the most isolated land mass in the world, and the only native land mammal in Hawai'i is an insectivorous bat (*Ziegler, 2002*. As a result, the flora and fauna in the islands evolved in the absence of mammalian predators and are naïve and lack defenses against them (*Tomich, 1969*; *Salo et al., 2007*; *Sih et al., 2010*; *VanderWerf, 2012*). With the arrival of Polynesians ~800 years ago (*Rieth et al., 2011*), the first land mammals were introduced, including the Pacific rat (*Rattus exulans*), domestic dog (*Canis familiaris*), and domestic pig (*Sus scrofa*; *Kirch, 1982*; *Burney et al., 2001*). Introduction of alien predators accelerated with the arrival of European colonizers in 1778, including the black or ship rat (*R rattus*) and Norway rat (*R norvegicus*), the domestic cat (*Felis catus*), small Indian mongoose (*Herpestes auropunctatus*), house mouse (*Mus musculus*), and European wild boar *(Sus scrofa)*. The set of invasive mammalian predators established in Hawai'i represents an existential threat to many native species, with ground-nesting birds and fruiting plants being the most vulnerable (*Ziegler, 2002*; *Lindsey et al., 2009*).

Fences capable of excluding all invasive mammals, including predators and herbivores, from juvenile mice all the way up to deer, were developed in the early 2000s in New Zealand, and dozens of such fences have been constructed in New Zealand and in Australia, protecting tens of thousands of hectares (*Legge et al., 2018*; *Innes et al., 2019*; *Innes et al., 2024*). These are often called "predator-proof fences" or "predator exclusion fences," though some species excluded are herbivores. They are tall enough to prevent large mammals from jumping over them, have a curved hood on top to prevent rodents and cats from climbing over, small mesh to prevent them from squeezing through, and a buried skirt to prevent digging underneath (*Day & MacGibbon, 2007*; *Young et al., 2012*; *Young et al., 2013*). After fence completion, the goal is to remove all mammalian pests from inside the fence, effectively resulting in the creation of "mainland islands" that have been shown

to provide more effective protection than ongoing predator control (*Long & Robley, 2004*; *Ringma et al., 2017*; *Bombaci, Pejchar & Innes, 2018*; *Binny et al., 2020*; *Pacioni, Kennedy & Ramsey, 2021*).

In 2011, the first predator exclusion fence in Hawaiʻi was completed at Kaʻena Point Natural Area Reserve, Oʻahu, and all mammalian predators were eradicated shortly after (*Young et al., 2012*; *Young et al., 2013*). Since then, 11 more predator exclusion fences have been completed in Hawaiʻi, with several more planned or under construction (*Young et al., 2018*; this article). Additional predator exclusion fences have been completed that were designed to exclude different subsets of invasive animals, such as ungulates and cats but not rats, or only rats and predatory snails to protect endemic tree snails and other species.

In this article we describe the predator exclusion fences that have been built in Hawaiʻi, including the efficacy and durability of various design features such as mesh type, post type, and fully enclosed *vs.* peninsula fences. We also assess the subsequent predator eradication attempts and the results achieved in protecting the target native species. We believe this information will allow managers to make informed decisions about future projects and the best fence designs for different circumstances.

## MATERIALS & METHODS

We compiled information about predator exclusion fences built and planned in Hawaiʻi by contacting landowners, natural resource managers, and fencing contractors. We included all fences designed to exclude all species of mammalian predators in Hawaiʻi in this analysis. We did not include fences designed to exclude only ungulates, only cats but not rats, and only rats but not cats, because the designs for such fences are quite different. We included fences that were actively under construction or had reached the final planning stages, but we excluded them from analyses of predator removal, monitoring outcomes, and maintenance, because those activities had not been completed yet.

### Fence design, purpose, and maintenance

We collected information on the fence itself (length, area enclosed, materials used, type, year completed, and location), on the project purpose (which species were protected), the habitat type protected, the funding source, and whether the project was for protection of existing resources or mitigation required under the United States Endangered Species Act (ESA), *i.e.,* to compensate for negative impacts to protected species that occurred or were expected to occur elsewhere.

We compiled information about maintenance issues related to fence materials or construction for all completed fences. This included construction defects, materials defects, design problems, corrosion, environmental impacts, and events such as extreme weather or tree fall. We examined the cumulative information to search for trends related to material type, construction, environment, or fence component.

### Predator removal and incursions

We collected information on the mammal species present at each site prior to fence construction, which species were targeted for removal, the methods used to remove them,

and whether each species was successfully eradicated. We considered a species to have been eradicated if it was not detected with at least two detection methods for more than four months. While this is shorter than the two-year standard used in larger island-wide eradications (*Parkes, Fisher & Forrester, 2011*), the small size of many of these sites allowed a higher detection probability compared to large sites. We also gathered information on mammal incursions that occurred after a species had been eradicated, including whether they were isolated, infrequent events, or ongoing and chronic (*i.e.* happening multiple times without appearing to establish a breeding population), and whether an incursion resulted in re-establishment of a breeding population (*i.e.* a reinvasion). We used six months as the definition of chronic because it typically takes 1–2 months to identify that an incursion has occurred, the source of the incursion, and another 2–4 months to determine if breeding occurred and to re-eradicate anything that had re-invaded.

## Monitoring of outcomes

Extensive biological monitoring was conducted within completed fences that protected existing populations of native species, primarily on the reproductive success of the target native bird species. We did not include two fences (Hiʻi and Kilauea Point National Wildlife Refuge; KPNWR) that were only completed in 2023, because outcome monitoring efforts were not yet complete.

Methods for monitoring wedge-tailed shearwaters (*Ardenna pacifica),* Laysan albatross (*Phoebastria immutabilis*), and other ecosystem parameters can be found in *Young et al. (2009)*, *VanderWerf et al. (2014)*, *Young et al. (2012)* and *Young et al. (in press)* and were used at both Kaʻena Point and Kuaokala. Briefly, this included monitoring of hatching, fledging, and reproductive success for Laysan albatross, and measuring overall reproductive success for wedge-tailed shearwaters. Methods used for monitoring waterbirds at Honouliuli can be found in *Christensen et al. (2021)*.

Of the completed fences where the purpose was mitigation to create a new seabird colony through either social attraction and/or translocation (*VanderWerf et al., 2023a*; *VanderWerf et al., 2023b*), four of seven had been erected long enough for results to have been achieved: Nihoku on Kauaʻi, two sites at Makamakaole on Maui, and James Campbell National Wildlife Refuge on Oʻahu. At the other three sites on Kauaʻi, social attraction had been conducted for only two years, which is not enough time to determine if the projects were successful because many seabirds do not breed until they are 3–5 years old. We excluded them from analysis for this reason.

## RESULTS

Twelve predator exclusion fences were constructed in the Hawaiian Islands from 2011–2023 and six more were either under construction or fully funded and scheduled to start construction in 2024 (Fig. 1); no other predator exclusion fences have been constructed in the United States outside of Hawaiʻi to date. Frequency of fence inspections varied among sites, but most often was quarterly (four times per year) or opportunistic. None of the fences used any remote surveillance systems to detect fence breaks.
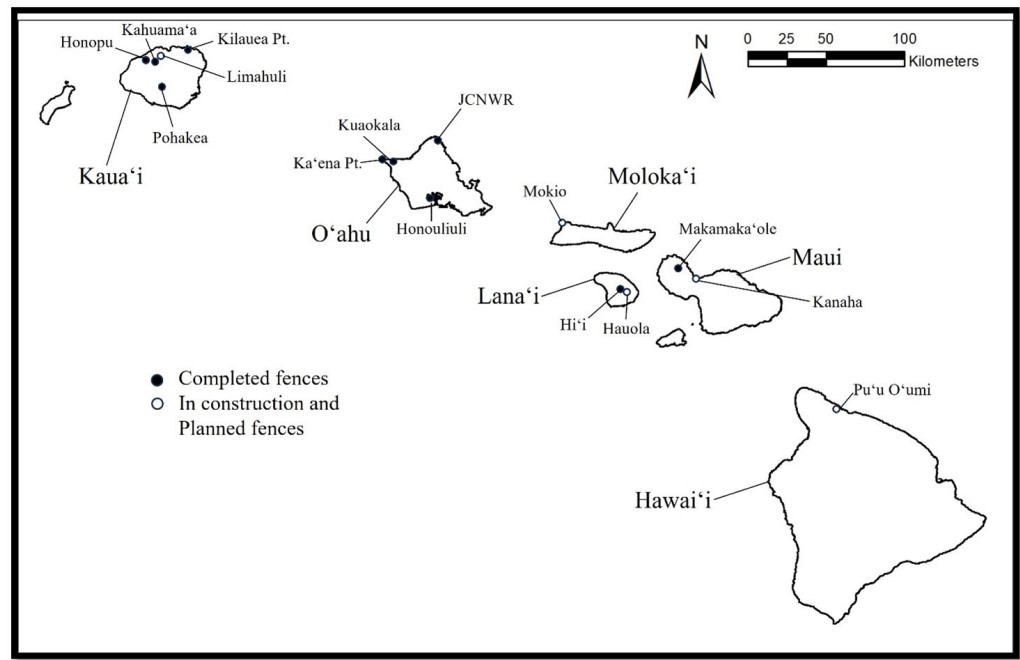

**Figure 1 Map of existing and planned predator exclusion fences in Hawai'i.** Black circles represent completed fences; white circles represent in-construction or planned fences. Image credit: Pacific Rim Conservation.

## Fence purpose and species protected

All 18 existing or planned full predator exclusion fences in Hawai'i were built to protect native birds, including 15 to protect seabirds and three for wetland birds (Table 1). Ten of the fences for seabirds were designed to protect Newell's shearwater (*Puffinus newelli*), Hawaiian petrel (*Pterodroma sandwichensis*), or band-rumped storm-petrel (*Oceanodroma castro*), which are the only three seabirds listed under the ESA in Hawai'i. Only one completed fence (Honouliuli) was built to protect endangered waterbirds, though two more planned fences are for waterbird protection (Kanaha Pond and James Campbell National Wildlife Refuge boundary fence). As of 2023, no fences had been built for the protection of endemic forest birds and only one fence, Ka'ena Point, had been built with recovery of native plants listed as a co-objective to protecting seabirds.

Twelve of the 18 fences (67%) were built to protect existing bird populations, and six fences 33%) were built for mitigation purposes to compensate for impacts that occurred or were expected to occur to species listed under the ESA. Of the six mitigation fences, all were for seabirds, of which five used social attraction and one used translocation; none of the mitigation fences protected existing bird populations.

Six of the 18 fences were peninsula-style with open ends and 12 of the fences were complete enclosures. The peninsula-style fences were either in wetland habitat or coastal shrub and all of them terminated at either a cliff face or steep, rocky shoreline that was intended to act as a natural, if semi-porous, barrier to predator ingress. There were no

**Table 1 Summary of basic information about existing and planned predator exclusion fences in Hawai'i.**

| Island | Location | Fence length (m) | Area enclosed (ha) | Year fence completed | Initiation event | Purpose | Goals achieved? | Land ownership | Funding source |
|---|---|---|---|---|---|---|---|---|---|
| Hawai'i | Pu'u O'umi | 500 | 1.0 | 2024[*] | Conservation | Seabird social attraction | Not complete | State | Private |
| Kaua'i | Honopu | 577.9 | 1.2 | 2022 | Mitigation | Seabird social attraction | Too early | State | Federal |
| Kaua'i | Kahuama'a | 866.2 | 3.6 | 2021 | Mitigation | Seabird social attraction | Too early | State | Private |
| Kaua'i | KPNWR | 3,200.2 | 101.2 | 2023 | Conservation | Existing seabirds | Too early | Federal | Federal |
| Kaua'i | Nihoku | 617.2 | 2.4 | 2014 | Mitigation | Seabird translocation | Yes | Federal | Private |
| Kaua'i | Pohakea | 304.8 | 1.2 | 2021 | Conservation | Seabird social attraction | Too early | State | Private |
| Kaua'i | Upper Limahuli | 853 | 2.8 | 2025[*] | Mitigation | Seabird social attraction | Not complete | Private | Private |
| Lana'i | Hauola | 482 | 1.0 | 2025[*] | Conservation | Seabird social attraction | Not complete | Private | Private |
| Lana'i | Hi'i | 2,910.7 | 32.4 | 2023 | Conservation | Existing seabirds | Too early | Private | Private |
| Maui | Kanaha | 4,506 | 57.9 | 2024[*] | Conservation | Existing waterbirds | Not complete | State | Multiple |
| Maui | Makamakaole A | 640.0 | 1.8 | 2013 | Mitigation | Seabird social attraction | yes | State | Private |
| Maui | Makamakaole B | 542.5 | 1.8 | 2013 | Mitigation | Seabird social attraction | No | State | Private |
| Moloka'i | Mokio | 1,753 | 36.4 | 2024[*] | Conservation | Seabird social attraction | Too early | Private | Private |
| O'ahu | Honouliuli | 1,099.7 | 14.6 | 2018 | Conservation | Existing waterbirds | Yes | Federal | Federal |
| O'ahu | JCNWR | 1,124.7 | 6.5 | 2015 | Conservation | Seabird translocation | Yes | Federal | Multiple |
| O'ahu | JCNWR boundary | 4,877 | 259.0 | 2025[*] | Conservation | Existing waterbirds | Not complete | Federal | Multiple |
| O'ahu | Ka'ena Point | 670.5 | 23.9 | 2011 | Conservation | Existing seabirds | Yes | State | Multiple |
| O'ahu | Kuaokala | 609.6 | 1.6 | 2021 | Conservation | Existing seabirds | Yes | State | Private |

**Notes.**

*Estimated date of completion for fences currently under construction.

peninsula-style fences in montane environments. Of the 12 fully enclosed fences, two were in coastal shrub and the remaining 10 were in montane environments above 500 m in elevation.

Of the 18 fences, 11 (61%) were exclusively funded by private foundations or landowners, four (22%) were a combination of private and federal funding, and three (17%) were entirely federally funded. Nine of the fences were on state land (50%), five (28%) were on federal land (all national wildlife refuges), and four (22%) were on private land.

## Fence design and materials

Completed fences ranged in length from 304–3,200 m and enclosed 1.2–101 ha. Planned fences ranged in length from 482–4,877 m in length and will enclose 2.5–640 ha. The design of the first predator exclusion fence at Kaʻena Point Natural Area Reserve (NAR) is similar to the design used in New Zealand (*Day & MacGibbon, 2007*; *Young et al., 2012*). The two Maui fences also used a variant of this design, but more recent fences used a variety of designs that differed in several ways, including mesh material and design, post material, and hood shape (Table 2). The overall design used for nine of the completed fences is shown in Fig. 2.

Fences in Hawaiʻi have used two mesh types (welded wire panels or rolls, and mini chain link rolls) and three mesh materials (stainless steel, galvanized steel, and PVC-coated galvanized steel). Fences built of welded wire consisted of two sections; an upper section that comprised most of the vertical fence, and a lower section that formed the underground skirt, which were joined with a horizontal metal bar. In fences composed of two mesh sections, the mesh type and material were the same for the upper and lower sections, but at Honouliuli and JCNWR, repairs to the lower section were made with a powder-coated stainless steel mesh. Twelve fences (67%) used a mini chain link design that came in 10 m rolls and could be contoured to sloping terrain. All but one of the mini chain link fences used 304 stainless steel as the material. Welded wire mesh was typically less expensive than mini chain link and could be constructed in panels, but the inflexibility made it difficult to work with in areas with slopes and thus was used in only six (33%) of the fences. Of the welded mesh fences, one was stainless steel, two were galvanized steel, and the remaining three (all planned fences) used a PVC-coated galvanized steel.

The three earliest fences in Hawaiʻi used the hood design from New Zealand shown in *Young et al. (2012)*, which extends diagonally downward from the top of the fence and has a curled lip on the outer edge. The 15 later fences (83%) used a modified design in which the hood sloped upward and then downward and had no curl at the outer edge (Fig. 2). Thirteen fences (72%) had a hood made of 304 stainless steel, four fences (22%) had a galvanized steel hood, and one fence had a powder-coated 304 stainless steel hood. The hood material was the same as the mesh material in most cases, but in the first fence at Kaʻena Point the mesh was 304 stainless steel and the hood was galvanized steel, and at three of the planned fences the mesh is PVC-coated galvanized steel and the hood is 304 stainless steel. The hood bracket material was always the same as that of the hood itself.

Post materials were either 13–16 cm diameter treated round wooden posts or five cm diameter stainless steel posts. Twelve fences (67%) used wooden posts exclusively, two

**Table 2  Summary of design aspects and materials used for all existing and planned predator exclusion fences in Hawai'i.** SS stands for stainless steel.

| Island | Location | Style | Environment | Hood material | Post material | Mesh type | Mesh material | Mixed metals? | Culvert? |
|---|---|---|---|---|---|---|---|---|---|
| Hawaii | Pu'u O 'Umi | Enclosed | Montane | 304 SS | 304 SS | Mini chain link | 304 SS | No | No |
| Kaua'i | Honopu | Enclosed | Montane | 304 SS | Wood | Mini chain link | 304 SS | No | No |
| Kaua'i | Kahuamaa | Enclosed | Montane | 304 SS | Wood + 304 SS | Mini chain link | 304 SS | No | No |
| Kaua'i | KPNWR | Peninsula | Coastal shrub | 304 SS | Wood + 304 SS | Mini chain link | 304 SS | No | No |
| Kaua'i | Nihoku | Enclosed | Coastal shrub | 304 SS | 304 SS | Mini chain link | 304 SS | No | Yes |
| Kaua'i | Pohakea | Enclosed | Montane | 304 SS | Wood | Mini chain link | 304 SS | No | No |
| Kaua'i | Upper Limahuli | Enclosed | Montane | 304 SS | Wood | Mini chain link | 304 SS | No | No |
| Lana'i | Hauola | Enclosed | Montane | 304 SS | Wood | Mini chain link | 304 SS | No | No |
| Lana'i | Hi'i | Enclosed | Montane | 304 SS | Wood | Mini chain link | 304 SS | No | No |
| Maui | Kanaha | Peninsula | Wetland | 304 SS powder coated | Wood | Welded | PVC-coated galvanized steel | Yes | No |
| Maui | Makamakaole A | Enclosed | Montane | Galvanized steel | Wood | Welded | Galvanized steel | No | No |
| Maui | Makamakaole B | Enclosed | Montane | Galvanized steel | Wood | Welded | Galvanized steel | No | No |
| Moloka'i | Mokio | Peninsula | Coastal cliff | 304 SS | Wood + 304 SS | Welded | PVC-coated galvanized steel | Yes | No |
| O'ahu | Honouliuli | Peninsula | Wetland | 304 SS | Wood | Mini chain link | 304 SS | No | No |
| O'ahu | JCNWR | Enclosed | Coastal shrub | 304 SS | Wood | Mini chain link | 304 SS | No | No |
| O'ahu | JCNWR boundary | Peninsula | Wetland, coastal shrub | 304 SS | Wood | Welded | PVC-coated galvanized steel | Yes | No |
| O'ahu | Ka'ena Point | Peninsula | Coastal shrub | Galvanized steel | Powder-coated galvanized steel | Welded | 304 SS | Yes | Yes |
| O'ahu | Kuaokala | Enclosed | Montane | Galvanized steel | Wood | Mini chain link | Galvanized steel | No | No |

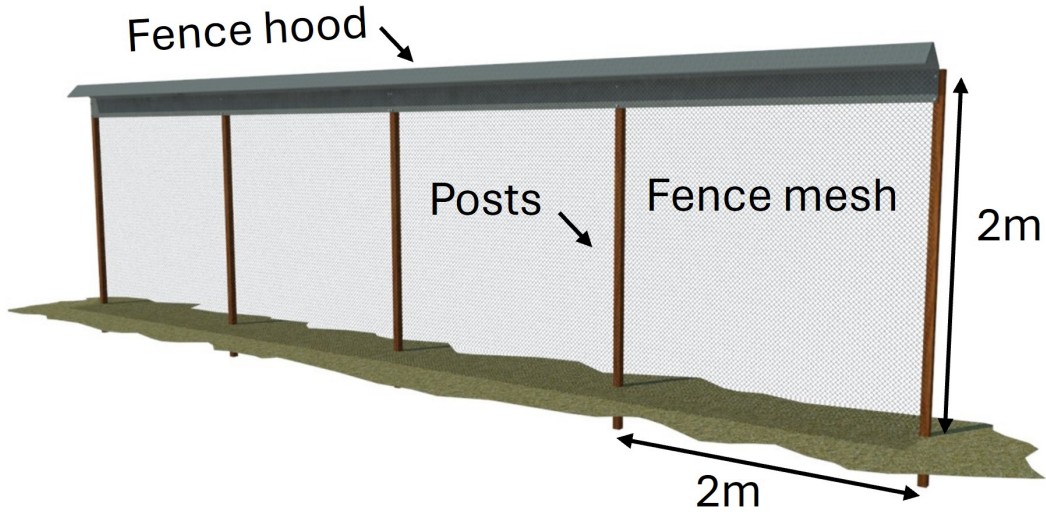

**Figure 2** Schematic of design used in most predator exclusion fences in Hawai'i.

(11%) used 304 stainless steel posts exclusively, three (17%) used a mix of metal and wooden posts depending on the specific terrain, and one used powder coated steel posts. Wooden posts were generally used in areas with soil substrates whereas metal posts were typically used in rocky areas.

Two of the completed fences (Ka'ena Point and Nihoku) each had a single culvert installed in ephemeral streams beds to allow for periodic water flow during periods of heavy rainfall. None of the existing fences crossed regularly flowing streams, and the fence lines were deliberately selected to avoid the use of culverts because they have been previously identified by several fence managers in New Zealand as the area with the highest potential for pest breaches. Culverts had a grate on the outside to prevent animals from entering, and at Nihoku, a grate was placed in the inside to prevent debris from entering the culvert since water flows from inside to outside the fence at that location. At two of the planned fences, Kanaha and JCNWR boundary, the fence will cross canals more than five m wide and up to several m deep and that have constant water flow, where it is not practical to use a culvert. At Kanaha, the fence is being built on existing bridges over the canals, and the steep, concrete banks of the canals are anticipated to limit entry by animals by swimming and climbing the banks. At the JCNWR boundary fence, the fence will extend a short distance into the canals, and intensive trapping will be done to intercept any animals that attempt to swim inside.

## Maintenance

Maintenance issues were reported at 11 of the 12 completed fences. Most maintenance issues arose within the first year after fence completion and almost all had arisen by the end of the second year, with the exception of an extreme rainfall event. The fence components most commonly requiring maintenance were the skirt (N = 6/11 fences, 55%) and the hood (N = 4/11 fences, 36%; Table 3).

**Table 3** Summary of maintenance issues in predator exclusion fences in Hawaii.

| Island | Location | Component | Material | Issue | Year fence completed | Year issue occurred | Solution |
|---|---|---|---|---|---|---|---|
| Kauaʻi | Honopu | Hood | 304 stainless | Treefall onto fence | 2022 | 2023 | Repair hood |
| Kauaʻi | Kahuamaʻa | Skirt | 304 stainless | Gaps under skirt from pig rooting | 2021 | 2022 | Cement and sod along skirt |
| Kauaʻi | Nihoku | Culvert | 304 stainless | Flooding from extreme rain event | 2014 | 2018 | Repair culvert |
| Kauaʻi | Pohakea | Gate | 304 stainless | Latch on gate didn't self-close | 2021 | 2021 | Replace latch |
| Lanaʻi | Hiʻi | Skirt | 304 stainless | Erosion on steep slope | 2023 | 2023 | Water bars to divert water |
| Maui | Makamakaole | Hood and skirt corrosion | Galvanized | Wrong material for environment | 2013 | 2015 | Replace corroded materials |
| Maui | Makamakaole | Hood and skirt corrosion | Galvanized | Wrong material for environment | 2013 | 2015 | Replace corroded materials |
| Oʻahu | Honouliuli | Skirt | 304 stainless | Acid in soil corroded skirt | 2018 | 2018 | Replace corroded materials |
| Oʻahu | JCNWR | Skirt, posts | 304 stainless | Acid in soil corroded skirt; ground termites and rot in posts, rodent digging | 2016 | 2018 | Replace corroded+rotted materials, extend skirt |
| Oʻahu | Kaʻena Point | Hood, rivets | Stainless mesh, galvanized hood, aluminum rivets | Electrolysis between metal types, metal fatigue | 2011 | 2012 | Replace corroded materials with correct metal type |
| Oʻahu | Kuaokala | Mesh | Galvanized | Wrong material for environment | 2021 | 2023 | Replace corroded materials |

Several of these issues were isolated events (tree fall, extreme rainfall events, defective gate latch) that were easily fixed, but several were chronic issues that resulted in large and expensive repairs, such as mesh failures along the entire fence. The most common preventable issue was using a less corrosion-resistant material (such as galvanized steel) instead of 304 stainless steel, which created the fence-wide repairs needed at Makamakaole and Kuaokala. The Kuaokala fence, despite being over 500 m in elevation in a montane environment and almost 1 km from the ocean, was still subjected to salt spray because of strong prevailing winds and steep terrain that forced winds upward. The use of galvanized steel rather than stainless steel resulted in complete failure of the mesh within two years of construction along the side of the fence facing the prevailing wind direction.

The next most common maintenance issue was related to soil type and chemical interactions with the skirt, which happened at JCNWR and Honouliuli NWR. At JCNWR, the southern boundary of the fence was built onto a limestone substrate, which dissolved in rainwater to produce carbonic acid that greatly accelerated corrosion of the stainless steel mesh wherever it touched the limestone, and this was exacerbated by rainwater running down the fence. The limestone substrate also contained numerous small holes and natural tunnels, some of which could have extended under the fence and allowed rodents to enter. On the northern boundary of the same fence, the soil consisted of sand with a variable

calcareous component, which also produced carbonic acid, but at a slower rate. This issue was largely corrected by adding cement over mesh sections that touched limestone or calcareous sand, though occasional small areas of corrosion still occurred on top of the cement. At Honouliuli on Oʻahu, part of the fence was located on fine, salty silt that when flooded became anaerobic mud, resulting in the production of hydrogen sulfide by anaerobic bacteria, which was converted to sulfuric acid at ground level through an interaction of sulfur, salt, and air. The corrosion at Honouliuli was particularly rapid and occurred within the first year of fence construction and required replacement of mesh in the sections with anaerobic mud.

At two fences, digging by animals resulted in fence breaches. At Kahuamaʻa, pig rooting along the edge of the skirt resulted in gaps that allowed rodents to enter. This was fixed by adding cement or sod over the edge of the skirt to fill the gaps and discourage rooting. At JCNWR, rodents occasionally dug tunnels under the skirt in the sandy soil. This was corrected by adding an extension to the skirt in the sandy area so it was 80 cm wide and extended 30–40 cm underground (instead of 30 cm wide and 10 cm underground), which required uncovering the skirt and then re-burying it after the extension was added.

Culverts resulted in maintenance issues at two sites. At Nihoku, an extreme rain event in 2018 resulted in accumulation of debris over the grate, which caused the ephemeral stream to flood over the culvert, causing erosion and gaps between the culvert and mesh. This event required excavating some debris and repositioning the culvert. At Kaʻena Point, the culvert across an ephemeral stream caused no issues, but there was a pre-existing older culvert from the 1940s that during fence construction was thought to be blocked, and closer inspection after the fence was completed showed it to have an opening that had allowed rodents to enter the fence. This issue was detected using tracking cards that showed a "hotspot" around the culvert, and it was corrected by installing a grate over that culvert too.

Another maintenance issue was mixing of metal types that caused accelerated corrosion by electrolysis. At Kaʻena Point, the use of a galvanized hood with stainless steel mesh and screws resulted in electrolysis that severely corroded the hood within two years. Several other interactions occurred when metal fasteners (*i.e.* nuts, bolts, rivets, and screws) of a different metal type were used. While the Mokio fence has not yet been completed, the fence has already experienced corrosion, particularly at contact points between the hood (which is 304 stainless steel) and cut sections of the PVC-coated steel (B. Haase, pers. comm., 2023). At Kaʻena Point, another issue related to metal type was widespread fatigue and failure of aluminum rivets used to secure two sections of mesh to each other. Repeated thermal expansion and contraction associated with daily heating and cooling caused the rivets to fatigue and become brittle, and similar daily expansion and contraction of the mesh sections they connected put strain on the rivets that they eventually could not withstand, causing the rivet heads to pop off.

## Predator eradication and incursions

Eleven of the 12 completed fences had completed eradications at the time of this publication; the KPNWR eradication was still underway and is not included in the results below. Cats

were present at all 11 sites and were successfully eradicated from all of them. Cats removed themselves from nine (82%) of the sites by climbing over the fence, which is possible because the hood is directed outward, and if they remained, they were removed with a combination of live trapping and shooting. The largest fence where cats self-exported was 32 ha at Hi'i on Lana'i.

Rats and mice were present in all 11 sites. All three species of rats were present in varying densities at each site with *R. rattus* being the most commonly detected species. Rats were targeted for removal at all sites and were successfully eradicated from eight of them (72%); successful projects used diphacinone bait stations placed 25–50 m apart and kill traps (*Young et al., 2013*; *Young et al., 2018*; *Christensen et al., 2021*, this article). Mice were targeted for removal in six of the 11 sites and were eradicated from five of those sites (83%) using the same methods as for rats. In 2023, the first aerial broadcast of rodenticide for rat eradication in Hawai'i took place within the Hi'i fence on Lana'i and appears to have been successful for both rats and mice despite only rats being targeted.

Mongooses were present in six of the 11 fence locations (mongoose are absent from the islands of Kaua'i and Lana'i); they were targeted and successfully eradicated at all sites with a combination of diphacinone bait stations, DoC 250 kill traps, and live trapping, depending on the site.

Pigs and axis deer (*Axis axis*) were each present at a single site, and they both were targeted for removal. Deer were removed from the Hi'i fence by targeted hunting. Pigs at KPNWR also are being removed by targeted hunting, and the eradication effort is still underway. It is too early to determine incursion rates for these species post-eradication.

Predator incursions occurred at all sites, but there was variation in the rate of incursions among predator species and among sites (Table 4). Two of the recently completed fences (Hi'i on Lana'i and KPWNR on Kaua'i) were excluded from incursion rate analysis because their eradications had just been completed or were still underway. There were no incursions by feral dogs at any site. There were a few cat incursions at peninsula fences, but none of the enclosed fences experienced a cat incursion and no sites have experienced a cat re-invasion, indicating that the fences are highly effective against cats.

Chronic mongoose incursions occurred at two sites, Honouliuli and Ka'ena Point. Mongoose incursion rates were low at Ka'ena Point, once every 217 days or about two per year, but persistent over time (*Young et al., 2013*). At Honouliuli, mongoose incursions were high at first, every 4–17 days (*Christensen et al., 2021*), when there was a corrosion problem with the fence skirt (see maintenance section above), but mongoose incursions stopped entirely after the problem was corrected by replacing the skirt. JCNWR experienced a single mongoose incursion of two individuals in 2019 during a period in which the skirt had holes due to corrosion; none have been detected inside the fence since the holes were patched. At Kuaokala, large scale failure of the mesh allowed at least three mongooses to enter the fence. At all sites that experienced mongoose incursions, the mongooses that entered the fence were trapped quickly, preventing reinvasion.

Rat incursions occurred in all fences. At fully-enclosed fences, incursion rates ranged from one rat in seven years (Nihoku) to 3.75 rats per year at the remaining four sites without reinvading. For the two peninsula-style fences with completed eradications, where

Young and VanderWerf (2024), *PeerJ*, DOI 10.7717/peerj.17694

**Table 4  Summary of predators present, removal techniques, eradication outcomes, and incursions rates for predator exclusion fences in Hawaii.** Rats represent three species of rats found throughout Hawaii: *Rattus rattus, R. exulans and R. norvegicus* with *R. rattus* being the most commonly detected species.

| Island | Location | Style | Predators present | Predators targeted | Eradication successful? | Eradication methods | Cats self-exported? | Incursions | Incursions chronic? | Cat incursion | Rat incursions | Mongoose incursions |
|---|---|---|---|---|---|---|---|---|---|---|---|---|
| Kauai | Honopu | Enclosed | Cats, rats, mice | All but mice | Cats | Bait stations, Traps | Yes | Yes | Yes | No | Not eradicated | N/A |
| Kauai | Kahuamaa | Enclosed | Cats, rats, mice | All | All species | Bait stations, Traps, Shooting | Yes | Yes | No | No | 1.5/year | N/A |
| Kauai | KPNWR | Peninsula | Cats, pigs, rats, mice | All but mice | Too soon | Bait stations, Traps, Shooting | No | Too soon | Too soon | Too soon | Too soon | N/A |
| Kauai | Nihoku | Enclosed | Cats, rats, mice | All | All species | Bait stations, Traps | Yes | Yes | No | No | 1/7 years | N/A |
| Kauai | Pohakea | Enclosed | Cats, rats, mice | All | All species | Bait stations | Yes | Yes | No | No | 1/year | N/A |
| Lanai | Hii | Enclosed | Deer, cats, rats, mice | All but mice | All target species | Aerial drop, Shooting | Yes | Too early | Too early | No | Too early | N/A |
| Maui | Makamakaole | Enclosed | Cats, mongoose, rats, mice | All but mice | Cats and mongoose | Bait stations, Traps | Yes | Yes | Yes | Unknown | Reinvasion | Regular; rate unknown |
| Maui | Makamakaole | Enclosed | Cats, mongoose, rats, mice | All but mice | Cats and mongoose | Bait stations, Traps | Yes | Yes | Yes | Unknown | Reinvasion | Regular; rate unknown |
| Oahu | Honouliuli | Peninsula | Cats, mongoose, rats, mice | All but mice | All target species | Bait stations, Traps | No | yes | yes | 1 in 2018 | 1/mo | 21–88/year, then 0 |
| Oahu | JCNWR | Enclosed | Dogs, cats, mongoose, rats, mice | All | All but mice | Bait stations, Traps, Shooting | Yes | Yes | Yes | No | 3.75/year | 1 incursion of 2 individuals |
| Oahu | Kaena Point | Peninsula | Cats, mongoose, rats, mice | All | All species | Bait stations, Traps | Yes | Yes | Yes | 1–2/year | 1/56 days | 2/year |
| Oahu | Kuaokala | Enclosed | Cats, mongoose, rats, mice | All | All species | Bait stations, Traps | Yes | Yes | Yes | No | Reinvasion | 3/3 years |

rat incursions are expected, incursion rates ranged from one rat every 12 days to one every 56 days. Mouse incursions occurred at all sites and were most frequent. At some sites it was not clear if mice were ever eradicated, and their nearly continuous presence made it impossible to measure incursion rates. Four of the fully enclosed fences experienced full reinvasions.

At the most recently completed fence at KPNWR, seven motion detection cameras (Ridgetek branded cameras using animl detection software) were deployed at fence ends, gates, and cliff tie ins in places where the biosecurity risk was deemed to be high. The images were automatically sorted by artificial intelligence to delete images without animals in them, and to draw boxes around animals detected in the photographs. Images with animals flagged in them were then emailed to staff in real time to facilitate a timeline response.

## Monitoring outcomes

In the completed fences that protected existing populations of native species (Ka'ena Point, Honouliuli, and Kuaokala), the response of native species to the removal of predators has been dramatic. At Ka'ena Point, reproductive success of wedge-tailed shearwaters rose from 0.29 young per nesting attempt before the fence to 0.50 after the fence, and the colony has grown dramatically in size (*VanderWerf et al., 2014*).

At Ka'ena Point and Kuaokala, hatching, fledging, and reproductive success of Laysan albatross were measured before and after fencing and subsequent predator removal. While hatching rates were comparable at both sites before and after predator exclusion fences and not significantly different, fledging rates increased by 25% from $0.60 \pm 0.09$ to $0.75 \pm 0.04$ chicks per pair per year ($df = 1$, N = 1,179; $X^2 = 31.03$, $p < .00001$) suggesting that chicks are more vulnerable to predation than eggs in this species. This resulted in an overall increase in reproductive success from $0.37 \pm 0.05$ to $0.43 \pm 0.03$ ($df = 1$, N = 1,984, $X^2 = 9.21$, $p = .0024$).

At Honouliuli NWR, *Christensen et al. (2021)* found that hatching success in Hawaiian stilts (*Himantopus mexicanus knudseni*) was significantly higher for pairs nesting inside the fence (0.83 hatching rate) than in pairs nesting in a nearby wetland with predator control but no predator exclusion fence (0.35 hatching rate). Additionally, a significantly higher number of eggs were laid per nest at the fenced site compared with the unfenced site. Together, this resulted in nearly three times the number of eggs hatched per nest inside the mammal-exclusion fence, compared with nests at the site with trapping alone.

Of the seven sites that were constructed for social attraction or translocation, sufficient time has elapsed since construction at four sites to evaluate whether they were successful. It should be noted that the success of social attraction is not related to the fence itself since birds are attracted to the sounds being played and decoys rather than the fence, but it is reported if it was listed as the purpose of building the fence. Of those four (two translocation and two social attraction) three have been successful in achieving their objectives of attracting birds to successfully breed at the site (*VanderWerf et al., 2019*; *Learned et al., 2023*; *Young et al., 2023*).

# DISCUSSION

Predator exclusion fences built in Hawai'i have been successful in permanently excluding deer, pigs, dogs, cats, and mongooses, and these fences have resulted in some dramatic responses by native species. In peninsula-style fences, there were occasional incursions of cats and mongooses around the fence ends, which was expected, but to date none of these have resulted in reinvasions of a fenced area. The fences were somewhat less effective at excluding rodents, but still resulted in substantial benefit to the resources targeted for protection. While rodent eradication was achieved in 72% of cases for rats and 83% of cases for mice, incursions occurred in every fence and 44% (N = 4/9) of those incursions resulted in reinvasions. Only two of the eight sites where rats were eradicated did not have chronic incursion issues, and thus rats remain, albeit at low densities, in 91% (10/11) of fences where they were targeted for removal. These numbers are very similar to findings of predator exclusion fences in New Zealand, where the vast majority of fences, including fully enclosed fences, have incursions (*Bell, 2014*). In New Zealand, peninsula fences have incursion rates ranging from 1–2 individuals per year for cats to 1–5 incursion events per year for rats (*Bell, 2014*).

The higher than hoped for incursion rates are likely due to a combination of factors related to the fence designs and local environmental conditions. It is clear that an enclosed fence made entirely of stainless-steel can keep out rodents at certain sites (Nihoku has gone many years without a single rodent detection for example), but there are occasional environmental interactions that can provide ingress points into some fences. The fence skirt appears to be the most vulnerable component and the one that has caused the most maintenance challenges. From ungulate rooting exposing the skirt edge on Kaua'i, to mud producing a metal-dissolving sulfuric acid at Honouliuli, to underground limestone tunnels at JCNWR on O'ahu that potentially allow rodent ingress, the issues have been varied and are different at each site. The commonality is that any hole under or through the skirt at ground level has been demonstrated to be vulnerable to and easily exploited by rodents. *Connolly, Day & King (2009)* found that 90% of rodents were able to exploit a fence hole at ground level within 6 h of its appearance, particularly if it occurred at night. Within 24 h of a hole appearing in a predator exclusion fence, there was a 99% chance of animals entering the fence. Thus, even with the most robust incursion monitoring and biosecurity program, it is virtually impossible to detect and respond to breaches in the time period required to prevent a rodent from entering.

The hood also was the cause of several maintenance issues, some of which were caused by the material and design. The modified hood designed in Hawai'i had several advantages over the original design from New Zealand used in the first several fences. It was simpler and thus cheaper to manufacture because it lacked the curled lip. It made the fence slightly taller and thus harder to jump over because the base of the hood sloped upward. Lastly, the curled lip on the original design was reported to be a liability because it was sometimes used by rodents as a travel corridor that facilitated their movement and collected water that accelerated corrosion.

Some maintenance issues that have occurred could be easily avoided, including corrosion from electrolysis among different metal types, the use of a lower grade metal, and unexpected interactions between the environment and soil. Future fence projects should aim to use the highest quality metal they can afford to prevent large scale failures. Soil testing prior to construction can be used to determine the potential for chemical interactions like those that occurred at JCNWR and Honouliuli. This can take the form of erecting several fence panels to determine how they weather in place, and/or sending the soil and metal for testing at an independent laboratory. If it is determined that the soil type could result in corrosion, metal posts should be avoided in that area, and alternative skirt materials should be explored.

The maintenance needs for each fence will be unique to the site and fence itself. Notably, half of the upcoming new projects will use a mix of materials (typically stainless steel hoods with a PVC-coated mesh) in an attempt to build larger fences with limited funding. While these materials hold promise, managers should ensure that mixed metal types are not in contact with one another to avoid electrolysis, and budget for maintenance needs. In addition, given that more than 50% of fences were built on state land but not funded by the state itself, funding streams should be identified for long term maintenance.

It is notable that none of the fences in Hawaiʻi have been constructed to protect forest birds, though a number of fences in New Zealand have been for forest birds (*Bombaci, Pejchar & Innes, 2018*; *Innes et al., 2024*). The most serious threat to most Hawaiian forest birds are diseases transmitted by introduced mosquitoes, but black rats are a serious threat to some Hawaiian forest birds, notably the Oʻahu elepaio (*Chasiempis ibidis*; *VanderWerf, 2009*; *VanderWerf, 2012*; *VanderWerf et al., 2023a*; *VanderWerf et al., 2023b*). Because seabirds use land only for nesting and not foraging, a small area can protect many nesting seabirds. While value exists for any sized predator fence to protect forest birds, a much larger fence would be required to protect forest bird populations rather than individual territories, particularly for territorial species, to provide enough food. Fences also have the potential to reduce pig populations, which are known to create mosquito breeding habitat in their wallows. Some forest birds thus are likely to benefit from predator exclusion fencing and this avenue should be pursued for future projects.

These fences also have enormous potential to protect threatened and endangered plant species from rodent predation, and indeed that was one of the primary justifications for the first predator exclusion fence project at Kaʻena Point. Several existing seabird fences have started endangered plant conservation in addition to seabird restoration activities, but to date there are not any fences exclusively for the protection of native plants.

A useful characteristic of seabirds is that they can be socially attracted to nest in new locations using acoustic and visual stimuli and thus protected areas can be created for them and then these techniques can be used to lure them in *VanderWerf et al. (2023a)*, *VanderWerf et al. (2023b)*. One-third of the fences in Hawaiʻi were built to facilitate creation of new seabird colonies, rather than existing ones. This approach comes with some risk and not all sites have resulted in the successful attraction of the target species. For sites using social attraction, doing a trial of the social attraction technique for the target species prior to construction may help determine if it is worth fencing the site, since social attraction

should be effective within 4–5 years (*Spatz et al., 2023*). If the target species is not actively prospecting or attempting to breed by that time, managers could consider investing in managing existing colonies, or moving towards translocation rather than creating a fenced area where birds may not colonize. Any trial of social attraction done prior to the construction of a fence should be done in conjunction with predator control. Predator fences are expected to last 25–30 years with regular maintenance. By constructing a social attraction fence for seabirds prior to establishing a breeding colony, managers are losing the benefits of 4–5 years colony protection based on the lifespan of the fence and the response time of the birds. In the worst-case scenario where birds do not come, the entire fence project may not have been necessary. Best practices for social attraction sites should also be followed to avoid constructing too many social attraction sites for the same species in a limited area and thus drawing from a limited pool of birds with the recommended distance between two social attraction sites being 100 km (*Gummer & Cotter, 2014*; *New Zealand Department of Conservation, 2014*; *Buxton et al., 2016*). There is limited contractor capacity to build and maintain fences in Hawaiʻi and thus site selection will be important to ensure that all fences can be maintained properly. These fences will be crucial for the long-term survival and protection of Newell's shearwaters and Hawaiian petrels in particular and should continue to be improved upon and expanded for the conservation of all endemic island taxa that are threatened by non-native mammalian predation.

## ACKNOWLEDGEMENTS

We thank the land managers at each site who made it possible for these fences to be constructed and to the kupuna whose land they reside on. We thank Gerry Kahookano in particular who has been instrumental in constructing many of these fences, and John Hinton, Xcluder, and Pono Pacific for construction of most of these fences, and Josiah Jury of Kuahiwi Wildlife Services for providing ongoing maintenance. We thank G2MT laboratories in Texas for conducting metal testing free of charge to inform metal selection at several sites. Finally, we thank our partners, the state of Hawaiʻi Department of Land and Natural Resources, The US Fish and Wildlife Services, Hallux Ecosystem Services, Maui Nui Seabird Recovery Project, Molokaʻi Land Trust, and Pulama Lanaʻi for sharing their experiences to help inform this study.

### Funding
This work was funded by the National Fish and Wildlife Foundation and the David and Lucile Packard Foundation. The funders had no role in study design, data collection and analysis, decision to publish, or preparation of the manuscript.

### Grant Disclosures
The following grant information was disclosed by the authors:
National Fish and Wildlife Foundation.
David and Lucile Packard Foundation.

## Competing Interests
The authors declare there are no competing interests.

## Author Contributions
- Lindsay Young conceived and designed the experiments, performed the experiments, analyzed the data, prepared figures and/or tables, authored or reviewed drafts of the article, and approved the final draft.
- Eric VanderWerf performed the experiments, prepared figures and/or tables, authored or reviewed drafts of the article, and approved the final draft.

## Data Availability
The raw data are available in the tables and are categorical.

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
