# Peer review of "A review of predator exclusion fencing to create mainland islands in Hawaiʻi"

_PeerJ, doi:10.7717/peerj.17694_

## Round 0.1 · original submission · Minor Revisions

Dear Dr. Young,

All three reviewers considered your manuscript to be very timely and positive. Therefore, I believe it will be accepted for publication after you proceed with the improvements indicated by the reviewers.

As soon as you prepare the new version, I will resend your text to the reviewers for a new evaluation and possibly its acceptance.

Sincerely,
Daniel Silva

·

Basic reporting

The paper presents a timely review of predator-proof fences in Hawai'i. It covers variation in their design, and experience with construction and maintenance. It also considers their effectiveness in meeting their conservation goals. The manuscript is well written in clear and professional English and is easy to read. It provides an extremely valuable review of this conservation tool and will be of great value to managers and others developing new such projects. Apart from a few minor corrections and comments, and perhaps 1-2 further references added (see suggestions in 4), I support publication of this paper.

Experimental design

The project gathered data from 18 Hawaiian fence projects either already in existence or in progress. These data are summarized effectively and reported on in the manuscript to provide clear results for each of the areas of investigation undertaken. One issue that wasn't reported on was variation in design, any maintenance problems, and effectiveness of the culverts within those projects that used them.

Validity of the findings

The focus is on predators, but some of the species excluded are herbivores or omnivores, e.g. goats, sheep, ungulates. Are you including herbivores amongst predators as a special case, or would another term be clearer than predator, e.g. mammalian pests? That is, are the herbivores excluded by these fences adequately included by the term 'predator-proof'.

I dispute the idea (line 45) that species on oceanic islands evolved without any predators. In New Zealand, for example, there were a range of avian predators that existed prior to invasion by mammalian predators. I recommend changing the sentence in the introduction that suggests that evolution occurred without predators to focus on evolution in the absence of mammalian predators only.

Can you explain ‘mitigation’ in more detail (as discussed in the Introduction). I assume this represents some form of compensation for reduction in populations caused by development on another site. As such, is this some form of biodiversity offsetting? Please clarify.

Common names should not normally start with a capital letter unless including a proper noun. This is inconsistent within the text.

Has there been any reincursions through the culvert at the sites that a culvert occurs?

How frequently were the fences inspected to detect holes or breaches in the projects surveyed? What types of surveillance systems on fence integrity were undertaken at the different projects?

Are there any results of monitoring the performance of endangered native plant populations inside fenced areas (as discussed lines 407-412)?

The last paragraph of the Discussion and the Conclusions section are almost identical. I suggest just providing this text once in the Conclusions.

Additional comments

Some mostly minor comments on the text are:

Line 30: Clarify that ‘…none of the mitigation fences protected existing populations of birds’ prior to fence installation.

Line 31: Similarly, were rats and mice present in every predator exclusion fence site prior to fence emplacement?

Line 47: ‘…to avoid or repel mammalian predators…’

Line 48: ‘…eradications of invasive mammals have been…’

Line 77: See this reference also for fence design:

Day, T. & MacGibbon, R. (2007) Multiple-species exclusion fencing and technology for mainland sites. Managing Vertebrate Invasive Species: Proceedings of an International Symposium (eds. G.W. Witmer, W.C. Pitt & K.A. Fagerstone), pp. 418 – 433. USDA/APHIS/WS, National Wildlife Research Centre, Fort Collins.

Line 80: ‘…effective than ongoing predator control’.

Line 86: ‘…mammals, such as ungulates and cats, but not rats,…’

Line 86: ‘predatory snails’ are not ‘invasive mammals’ as suggested earlier in this sentence.

Line 101: ‘…summary, but these were excluded…’

Line 117: For mammalian species present at each site, was this prior to fence instalment? Clarify.

Line 126: Suggest: ‘…to re-eradicate anything that had re-invaded.’

Line 133 and 137. Laysan albatross spelt different ways.

Line 140: Cite VanderWerf et al 2023 here and define what you mean by social attraction and translocation the first time it is mentioned. Miskelly et al. 2009 Biological Conservation would be another possible reference for seabird translocation methods.

Line 158: ‘sandwichensis’ misspelt

Line 187: Again, see Day and MacGibbon 2007 for a labelled diagram of this fence design, i.e. the Kiwi fence design.

Line 201: Please point out the major difference with these two different hood designs?

Line 210: Please describe how these culverts were designed to prevent or reduce pest incursions.

Line 214: Do you have a citation or pers. comm. to support this statement about the opinion of New Zealand managers? Also, managers of what?

Line 235: Two full stops at end of sentence.

Line 262: ‘heating’ misspelt.

Line 264: ‘causing the rivet heads to pop off.’

Line 272: Clarify how animals can get out of the predator-proof exclosure.

Line 288: Is it Axis deer or axis deer?

Line 321: What is the unit of reproductive success used here? 0.29 what? Give this unit where other reproductive success measures are used elsewhere in this section.

Line 399: Clarify that the forest birds you consider here are ‘forest birds in Hawai’i ‘. For effects of fences on native forest birds, a useful comparative reference is Bombaci et al. 2018 Ecosphere 9(11): e02497.

Line 407: Separate discussion on plants into a new paragraph.

Line 470: Change ‘the’ to ‘they’

Line 567: Don’t use parentheses for date in this reference.

Tables probably need further formatting before suitable for the journal.

Fig. 2 would be improved with some labels to identify the parts.

·

Basic reporting

This is a valuable, generally well-written account about predator exclusion fencing in Hawaii that methodically explains why and how they were built and what outcomes have been. It is clear that fence applications in Hawaii have some different and some same issues to those in NZ, so there is international value in this paper. There are no major problems with structure or content. The introduction concisely explains the mammal pest problem and the origins of pest fences. The sample of fences included is explained clearly (basically all of them with multi-species exclusion) and is substantial. The flow of the article is good and the discussion arrives at a very valuable and practical conclusion that seabird restoration and fence material selection could be trialled during planning phases. Tables are good BUT captions need to be expanded to explain details eg in Table 1 clarify that Length (m) is total fence length (not site length); Year completed is year the fence construction was completed (not eradication etc); Target species can be worked out from text but tables and figures should be self-contained. In Table 2, replace 'future' in caption with 'planned'. I personally think that 'Habitat' is wrong (it is a species-specific concept) and it should probably be 'Environment'. In table 4 (and throughout whole ms) can the rats not be named to species? If not, say so early and give the reason. Ship and Norway rats have different size, behaviours and impacts and sometimes control so please acknowledge this. In Table 4, redefine 'chronic' and perhaps 'incursion' cf 'reinvasion' so that Table is self-contained. Clarify for readers the meaning of 'incursion' in Table 5. Neither Table 2 nor 5 are referred to in the text. I suspect that 'Table 3' on line 297 should be 5? Figure 1 is good; figure 2 would benefit with having a scale on it; even a drawn human and cat might do.
Some other comments:
1. All citations except Salo et al. 2007 (line 61) are in the references.
2. There are more recent NZ collations of and comments on fenced sanctuaries than those cited eg Innes J, Fitzgerald N, Binny R, Byrom A, Pech R, Watts C, Gillies C, Maitland M, Campbell-Hunt C, Burns B. 2019. New Zealand ecosanctuaries: types, attributes and outcomes. Journal of the Royal Society NZ 49: 370-393, and Innes J, Norbury G, Samaniego A, Walker S, Wilson D. 2024. Rodent management in Aotearoa New Zealand: approaches and challenges to landscape-scale control. Integrative Zoology 2024: 19, 8-26.Doi 10.1111/1749−4877.12719 and (reviewing outcomes) Binny RN, Innes J, Fitzgerald N, Pech R, James A, Price R, Gillies C, Byrom AE. 2020. Long-term biodiversity trajectories for pest-managed ecological restorations: eradication versus suppression. Ecological Monographs 91(2), 2021, e01439. and Bombaci S, Pejchar L, Innes J. 2018. Fenced sanctuaries deliver conservation benefits for common and globally threatened native island birds in New Zealand. Ecosphere 9(11) Article e02497 and Bombaci S, Goldstein L, Flaherty T, Kelly D, Innes J. 2021. Excluding predators increases bird densities and seed dispersal in fenced ecosanctuaries. Ecology 102(6). e03340.
3. LIne 56, reduce should be reduces
4. Line 136 and elsewhere, you shouldn't cite an In review paper, rather refer to Unpub. data etc.
5. Line 231 delate space after skirt
6. Line 235. Two full stops after fence (there are more such editing things that I won't dwell on but need checking)
7. LIne 317, Replace 'Of the' with 'Inside'
8. 'Reproductive success' needs a clarificatory unit after 0.29. Does that mean that at least 29% of attempts? pairs? fledged chicks? laid eggs? etc

Experimental design

This is original primary research within the aims and scope of the journal. The introduction explains clearly why this work was done and the methods are also clear. I don't know that such a paper has been done anywhere else - we have not had a recent similar review like this in New Zealand where the fences originated.

Validity of the findings

A lot of the original data is presented in the Tables and they do not demand complex statistics. The conclusions arise logically from the review itself and are well stated and are well linked to stated intentions of the work (lines 88-93).

Additional comments

No additional comments.

Reviewer 3 ·

Basic reporting

This is an interesting and important review of exclusion fencing in Hawaii. It is generally well written and informative. I do think it would benefit from placing more soundly in the context of the international experience and literature – especially from Australia and New Zealand. I recommend this not simply as a matter of principle, but because I think it would help maximise the effectiveness of future predator-exclusion fences in Hawaii.
General points/comments
(1) I think the review could and should draw more on the international experience. The New Zealand (NZ) experience is particularly relevant, but also Australia. I don’t know the extent to which there has been interaction with practitioners in those two countries by fence builders in Hawaii, but many of the issues reported have been encountered by the former and could potentially have been avoided. I think this makes placing of this review in the international experience/literature particularly important because it will help avoid such issues for future fenced sanctuaries in Hawaii. Also, I think a key purpose of this paper is to put Hawaii on the map in terms of being a contributor to the practice and research on predator-proof fences internationally. A list of some useful papers includes the following, but I recommend a further search via Google Scholar or similar:
Innes, J.G., Norbury, G., Samaniego, A., Walker, S. and Wilson, D.J., 2024. Rodent management in Aotearoa New Zealand: approaches and challenges to landscape‐scale control. Integrative Zoology, 19(1), pp.8-26.
Innes, J., Fitzgerald, N., Binny, R., Byrom, A., Pech, R., Watts, C., Gillies, C., Maitland, M., Campbell-Hunt, C. and Burns, B., 2019. New Zealand ecosanctuaries: types, attributes and outcomes. Journal of the Royal society of New Zealand, 49(3), pp.370-393.
Legge, S., Woinarski, J.C., Burbidge, A.A., Palmer, R., Ringma, J., Radford, J.Q., Mitchell, N., Bode, M., Wintle, B., Baseler, M. and Bentley, J., 2018. Havens for threatened Australian mammals: the contributions of fenced areas and offshore islands to the protection of mammal species susceptible to introduced predators. Wildlife Research, 45(7), pp.627-644.
Long, K. and Robley, A., 2004. Cost Effective Feral Animal Exclusion Fencing for Areas of High Conservation Value in Australia: A Report. Victoria Department of Sustainability and Environment.
Moseby, K.E. and Read, J.L., 2006. The efficacy of feral cat, fox and rabbit exclusion fence designs for threatened species protection. Biological Conservation, 127(4), pp.429-437.
(2) I am not convinced, based on the NZ experience, that predator-proof fences would not have huge benefits for forest birds in Hawaii. I recommend re-visiting this in light of the NZ experience – see the NZ references above and also:

Innes, J., Kelly, D., Overton, J.M. and Gillies, C., 2010. Predation and other factors currently limiting New Zealand forest birds. New Zealand journal of ecology, 34(1), p.86.
While many birds may nest off the ground, rats, mice, cats and mustelids can climb trees, and I suspect excluding them (or reducing them in forests) with fences and feral control would yield increase in forest bird population productivity.
(3) Naming conventions. I am not sure of the journal policy on this, but there need to be a consistent use of naming conventions. Usually, for common names these are lower case unless the name is at the start of a sentence or contains the name of something.

(4) Utilize. The word utilize is used in a number of places throughout the manuscript. The simpler word “use” or “used” is better.




Detailed comments

Abstract
Line 5 – suggest removing “that was developed”.
Lines 13 – 15 – I am not clear on the difference between these two.
Line 17 - change “for” to “to support a”.
Line 19 – comma after “targeted”.
Line 21 – remove “were”, add “achieved” after “typically”.
Line 25 – comma after “colonies”.
Introduction
Line 53 – These NZ references are a bit outdated. I suggest keeping these, but also find some newer ones.
Line 59-61 – very similar to NZ – I encourage using the NZ experience/literature as context throughout the paper.
Line 63 and 68 – Sus scrofa – appears twice. Same species – though one is domesticated version, one wild.
Line 71 – 73 – need more references here and need to also talk about Australia. I have provided some examples of references.
Line 75 – “rolled hood” – please explain for readers who do not have a background in this area.
Line 86 – “cats but not rats” – why?
Line 125 – please use “because” rather than “as” when the former is what is meant.
Line 131 – acronyms need to be explained when first used.
Line 133 – see above on naming conventions.
Line 136 – see above on acronyms.
Line 138 – see above.
Line 139 - 2021 should be in parenthesis, not the full citation.
Line 140 – please explain “social attraction” to readers who are not familiar with this and or why it is being used here. This is assumed knowledge – remember to write for a scientifically educated lay audience, who may not know the background to a particular subject area.
Results
Line 158 – see above.
Line 159 – see above.
Line 166 – “future take under the ESA”. Not clear what is meant here?
Line 167 – 1-9 please spell out. Ten onwards can be a number.
Line 185 – space between number and unit.
Line 188 – I suggest “use” rather than “utilize” – see above.
Line 190 to 195 – what was the aperture size? Critically important to know this because it impacts what species we can expect to exclude.
Line 210 – I presume culverts had mesh/wire over them?
Line 217 – comma after “completion”.
Line 227- “because” rather than “as”.
Line 231 – remove space after “skirt”
Line 235 – two full stops.
Line 262 – “heating”?
Line 280 – five.
Line 288 – “axis”? – see above.
Line 289 – comma after “each”
Line 307 – “one”.
Line 312 - “because” rather than “as”.
Line 327 – check tense of manuscript. I suggest “were” rather than “are”
Line 331 – “stilts”?
Line 337 – Do you mean “insufficient”?
Discussion
Line 347 – perhaps change “fences” to “exclusion areas”?
Line 358 – Bell 2014 – an important reference, but also dated. I suggest adding some more updated references.
Line 378 – 390 – This is an important section, but as mentioned above, I think it would be good to place in the context of the NZ/Australian experience because this will likely save time and money for projects in Hawaii.
Line 393 – I think PVC-coated mesh is likely to degrade quickly (at least the coating).
Line 398 – 412 – please see my comments above. Based on the NZ experience, I suspect that predator-proof fences would be highly beneficial to forest birds (and other forest fauna).
Conclusion
This is a repetition of the paragraph above.

Experimental design

See 1.

Validity of the findings

See 1.

---

## Round 0.2 · Minor Revisions

Dear Dr. young,

All three had positive impressions regarding the improved version of your manuscript. Still, minor improvements are necessary before the manuscript is accepted for publication in PeerJ.

Sincerely,
Daniel Silva

·

Basic reporting

Thank you to the authors for their attention to the comments and suggestions of the reviewers. In my initial review I had confirmed that this paper filled an important gap in information and analysis on conservation fences in Hawaii and would be highly valuable. The revision provided improves on what was already a professionally formatted and well-written article. I found a small number of minor typographical corrections as listed below. Apart from these corrections, I support the publication of this paper.

Experimental design

No comment. Those issues identified in my previous review have been resolved to my satisfaction.

Validity of the findings

Again, those issues identified in my earlier review have been resolved to my satisfaction.

Additional comments

Some minor editorial corrections and suggestions identified on the revision are listed here.

Line 19-20: ‘…documenting the size and design of each fence, outcomes of predator eradications, maintenance issues at each fence, and the resulting native species responses.’
Line 66: Give the scientific name of the European wild boar, even though it is Sus scrofa again.
Line 87: ‘…such as ungulates and cats but not rats…’
Line 110: Delete ‘…for take…’
Line 218: Delete ‘…was the same…’
Line 224: Delete ‘…and…’
Line 347: Change ‘ranging’ to ‘ranged’.
Line 368: Change Albatross to albatross.
Lines 372-373: Include the unit for reproductive success here. Is this fledglings per female per year?
Line 414: Suggest rewording to ‘has been demonstrated to be vulnerable to and easily exploited by rodents’.
Lines 474-475: Suggest ‘managers are losing the benefits of 4-5 years of colony protection based…’
Line 489: Change ‘the’ to ‘they’
Line 505: ‘Cassey P’
Table 4: Note that it is Rattus norvegicus not R Norvegicus.

·

Basic reporting

I am happy with the authors' responses to all reviewers, so no further comment

Experimental design

No further comment, fine now.

Validity of the findings

No further comment, fine now.

Reviewer 3 ·

Basic reporting

The manuscript is much improved. I just have one note regarding the discussion of the value of predator-proof fences for forest birds in Hawai'i.

Forest birds. Apologies if I was not clear. I mean that predator-proof fences could be hugely beneficial to forest birds. In the section starting on page 398 in the original manuscript, the authors seemed a little dismissive of the value of fences for forest birds when compared to sea birds (though I note they did say that it is an avenue that should be pursued in future).

I think the evidence of the value of fences for forest birds in New Zealand is unequivocal. I was concerned that by saying fences for seabird are more relatively important/higher conservation value, this could be misinterpreted. The decision to build a fence depends on the aims of the project and the values and resources of those behind them.

Further, in terms of the size of fenced-areas – forest birds do not necessarily need all their home range/life history to be fulfilled within the fence if there is threat control in the surrounding landscape. There are now many examples of bird species in New Zealand spilling over into surrounding habitat outside predator-proof fences. I don’t think that fence options should be framed as competing choices. The revision of this section is much improved, but I encourage the authors to consider the framing a little more.

Experimental design

n/a

Validity of the findings

n/a

Additional comments

See above.

---

## Round 0.3 · accepted · Accept

Dear Dr. Young,

I am pleased to accept your manuscript for publication in PeerJ.

Sincerely,
Daniel Silva